# MarkQA: A large scale KBQA dataset with numerical reasoning

**Xiang Huang, Sitao Cheng, Yuheng Bao, Shanshan Huang, Yuzhong Qu**

State Key Laboratory for Novel Software Technology, Nanjing University, China

{xianghuang, stcheng, yhbao, shanshan_huang}@smail.nju.edu.cn, yzqu@nju.edu.cn

## Abstract

While question answering over knowledge bases (KBQA) has shown progress in addressing factoid questions, KBQA with numerical reasoning remains relatively unexplored. In this paper, we focus on the complex numerical reasoning in KBQA and propose a new task, NR-KBQA, which necessitates the ability to perform both multi-hop reasoning and numerical reasoning. We design a logic form in Python format called PyQL to represent the reasoning process of numerical reasoning questions. To facilitate the development of NR-KBQA, we present a large dataset called MarkQA, which is automatically constructed from a small set of seeds. Each question in MarkQA is equipped with its corresponding SPARQL query, alongside the step-by-step reasoning process in the QDMR format and PyQL program. Experimental results of some state-of-the-art QA methods on the MarkQA show that complex numerical reasoning in KBQA faces great challenges.

## 1 Introduction

Knowledge-based question answering (KBQA) aims to answer a question over a knowledge base (KB). It has emerged as a user-friendly solution to access the massive structured knowledge in KBs (Lan et al., 2021; Shu et al., 2022). Among the extensive knowledge stored in KBs, the quantitative property is the 3rd most popular property on Wikidata (only less than WikibaseItem and ExternalID), and there are more than 95M facts with quantitive properties. Given the large amount of quantitative facts stored in KB and the ability to perform precise symbolic reasoning through query languages, it is natural to use KBQA as a solution to real-world problems that require numerical reasoning.

However, it is shown that existing KBQA datasets are insufficient for numerical reasoning. We find that only 10% and 16.2% of questions in ComplexWebQuestions(CWQ) (Talmor and Be-

rant, 2018) and GrailQA (Gu et al., 2021), respectively, require numerical reasoning, but this part of the questions just focuses on some aggregation operations and lacks complex multi-step numerical reasoning. The remaining only need to match a graph pattern on KB (multi-hop reasoning), without the need to perform numerical reasoning. As a result, the questions that require complex numerical reasoning has not been covered by previous datasets (e.g. "How much more VAT do you have to pay to buy the most expensive iPhone 13 in Russia than in Japan?" or "How many times longer is the longest aircraft carrier than the shortest?").

In this paper, we propose a new challenging task, **NR-KBQA** (Knowledge-based Question Answering with Numerical Reasoning). Different from traditional KBQA which mainly focuses on multi-hop reasoning, NR-KBQA focuses on numerical reasoning and its combination with multi-hop reasoning. As shown in the left part of Figure 1, multi-hop reasoning needs to match a graph pattern in the KB, the difficulty of which comes from the composition of KB items (entities, relations, and classes). On the other hand, the difficulty of numerical reasoning comes from the composition of mathematical operators. It is worth noting that computational tasks entail the involvement of multiple operands, while each operand may be obtained by matching a graph pattern on the KB. Consequently, the combination of multi-hop reasoning and numerical reasoning will lead to a combinatorial explosion of the number of logical form structures, which poses a huge obstacle to semantic parsing models.

To support the study of this task, we construct a large-scale dataset called **MarkQA** (CoMplex NumericAl Reasoning over Knowledge Base Question Answering Dataset) which starts from 1K questions posed by humans and automatically scales to 32K examples. Each question in MarkQA is equipped with a question decomposition in the widely used QDMR format and our pro-

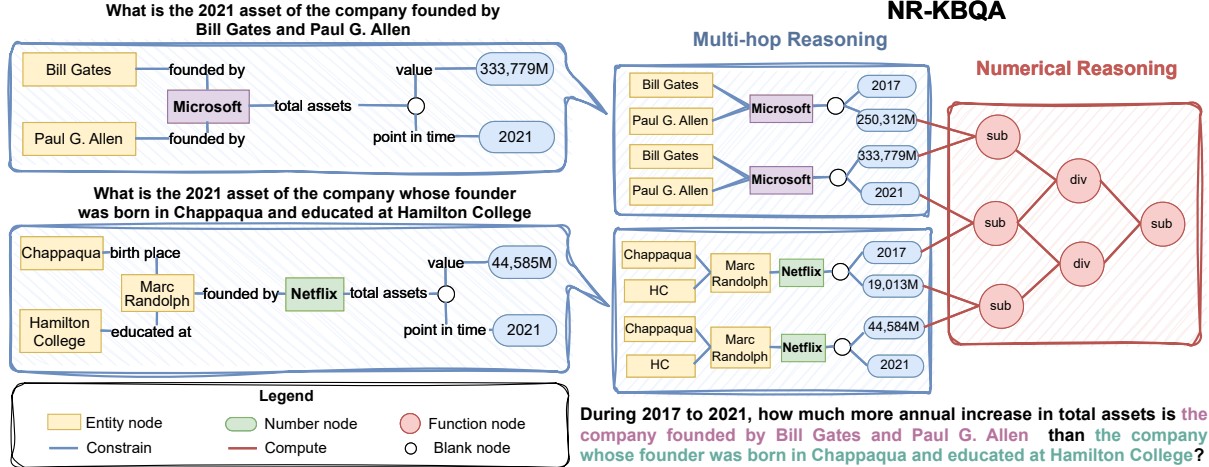

Figure 1: An example shows multi-hop reasoning, numerical reasoning, and their combination.

posed corresponding logic form in Python, namely PyQL. The QDMR can be seen as an explicit reasoning path and regarded as a question decomposition resource. Meanwhile, PyQL not only acts as a reasoning step but can also be directly transformed into a SPARQL query. It offers a more human-readable alternative to SPARQL and can easily be extended for further developments. We believe that MarkQA will serve as a valuable resource to foster further development of KBQA. [1]

The remainder of this paper is organized as follows: Section 2 reviews related work. Section 3 defines the NR-KBQA and introduces PyQL. The construction and analysis of MarkQA are demonstrated in Section 4. Section 5 presents experimental results. We conclude our contributions and future work in Section 6.

## 2 Related work

### 2.1 Knowledge-based question answering

KBQA refers to QA systems that take KB as the underlying knowledge source. The primary focus of KBQA research is on identifying graph patterns in the KB that satisfy the constraints (multi-hop reasoning), leaving numerical reasoning uncovered.

This influences the construction of KBQA datasets. Most KBQA datasets (Trivedi et al., 2017; Talmor and Berant, 2018; Gu et al., 2021) are essentially built through the OVERNIGHT (Wang et al., 2015) method or its extensions, consisting of three steps: sample logic forms(LF), convert LF to canonical questions, and paraphrase them into natural language questions (NLQ) through crowd-

sourcing. We refer to this manner as LF-to-NLQ. Though efficient, they only consider sampling LFs from a single connected sub-graph in KB. Consequently, the generated question is mostly a factoid question aiming to find an entity or entity set that meets the specific conditions, thereby only realizing multi-hop reasoning. However, many real-world questions require the introduction of complex numerical reasoning over multiple connected subgraphs, which can not be covered by the previous construction methods. When multi-hop reasoning and numerical reasoning intertwine, it is insurmountable to choose reasonable query patterns by enumerating all possible ones, given that the search space can comprise millions of patterns, of which only a minuscule fraction is deemed reasonable.

To ease this problem, we collect some questions as seeds in an NLQ-to-LF manner, making sure the questions are reasonable. Meanwhile, we propose a general framework called **S**eeds-t**o**- **F**orest (SoF) to automatically scale the dataset.

### 2.2 Numerical Reasoning

There are several types of QA tasks involving numerical reasoning:

1) Some KBQA datasets (Talmor and Berant, 2018; Gu et al., 2021) consider counting, superlatives (argmax, argmin), and comparatives (>, >=, <, <=). But their questions only perform the calculation at most once, and the types of operator are limited to just comparison and some aggregations.

2) The most famous Machine Reading Comprehension (MRC) dataset with numerical reasoning is DROP (Dua et al., 2019). It also lacks complex numerical reasoning. As a result, a large por-

---

[1] https://github.com/cdhx/MarkQA

tion of methods of DROP follow a Multi-predictor-based manner (Dua et al., 2019; Hu et al., 2019) which employs multiple predictors to derive different types of answers. For example, they use a multi-classifier to find the answer to a counting question as the answer is restricted to integers from 0 to 10. This is a far cry from actually solving a numerical reasoning problem.

3) Math Word Problem (MWP), such as MATHQA (Amini et al., 2019), ASDiv (Miao et al., 2020), and LILA (Mishra et al., 2022), require intricate complex numerical reasoning. However, MWP is based on a virtual scenario, while real-world problems often require to access accurate knowledge. MWP model does not require the ability to query knowledge from knowledge sources, but just the ability to correctly understand the question itself and reason numerically.

4) Table QA, such as FinQA (Chen et al., 2021), may involve multi-step numerical reasoning. Compared to FinQA, we focus on KBQA tasks in general rather than the financial domain. Besides, compared to Table QA, KBQA has some inherent superiority in terms of the amount of knowledge, aggregation for large search space, and ease of multi-hop multi-domain querying.

To alleviate the above problems, our dataset requires not only the ability to interact with underlying knowledge sources but also the ability to perform complex numerical reasoning.

## 2.3 Interpretable Reasoning Path

When it comes to complex reasoning (e.g., multi-hop, numerical, or logical reasoning), it is crucial to examine whether a model really deeply understands the underlying problem-solving process rather than merely producing the answers.

There are mainly two common representations of reasoning path: question decomposition in natural language, and formal representation in symbolic language. QDMR (Wolfson et al., 2020) is a widely used question decomposition format. It decomposes a question into a sequence of reasoning steps, and each step is an intermediate question phrased in natural language. Some works instead choose a more formal symbolic language representation. LILA (Mishra et al., 2022) and Binder (Cheng et al., 2023) adopt Python to showcase the process of reaching the answer. (Cao et al., 2022) develops a Knowledge oriented Programing Language (KoPL). Different from KoPL, our proposed PyQL considers more comprehensively the support of numerical reasoning and more complex SPARQL grammars. Moreover, while KoPL requires the development of a specific execution engine and can only execute on a JSON format KB, PyQL can directly be compiled to executable SPARQL queries, without the burdensome design of an additional executor and supporting native KB format.

In MarkQA, we provide each question with a QDMR and a PyQL to present the reasoning process. We believe it would be a valuable resource to improve the interpretability of the AI system.

## 3 NR-KBQA

In this section, we present the formal definition of a question with numerical reasoning (NRQ), which is the base of NR-KBQA. Then, we introduce PyQL to showcase the reasoning steps of NRQ.

### 3.1 Numerical Reasoning Question

An NRQ is any question, requiring mathematical calculations, such as arithmetic calculation, aggregation, or comparison, to reach the answer. An NRQ essentially consists of the descriptions of value and the computation process. The connotation of NRQ can be defined recursively in Backus–Naur form:

$$\text{<NRQ> ::= <Func> <Arg> \{ <Arg> \}} \quad (1)$$

$$\text{<Arg> ::= Num | <NRQ> | <Des>} \quad (2)$$

$$\text{<Des> ::= Rel  | <Des> <Des>} \quad (3)$$

$$\text{ ::= Ent | Num | <Des> | <NRQ>} \quad (4)$$

In this grammar, <NRQ> represents the intrinsic meaning of NRQ, which can be regarded as the outermost function (<Func>) applied to one or more <Arg>. <Arg> corresponds to a constant value (Num), a description of an entity's numerical attribute (<Des>), or another <NRQ>. <Des> describes the relationship (Rel) between a variable () and the entity being described, while the  corresponds to an entity (Ent), a Num, a <Des> or a <NRQ>. Equation 2 and 4 allow for the nesting of numerical and multi-hop reasoning, thereby enabling the representation of complex NRQ.

Based on this recursive nature, the query graph of an NRQ can be modeled as a tree and that of a sub-question is a sub-tree. For the example in Figure 1, the right part (in red) is a computational tree where each intermediate node is a function

node and each leaf is a constant value or an attribute value (green node). The attribute value node is acquired by matching a graph pattern in the KB, which the previous datasets focus on and can be seen as a description of a value.

## 3.2 PyQL

We propose PyQL (**Py**thonic **Q**uery **L**anguage for SPAR**QL**), a logical form written in Python as a reasoning step representation for NRQ. A PyQL is a sequence of commands: $\{c_1, c_2, ..., c_n\}$, where $c_i$ either initializes a PyQL object or calls a function on the object. As shown in the top left of Figure 2, the user should first initialize a PyQL object and sequentially add functions to construct the whole query. Each function represents a reasoning step such as stating the relation between two entities or computing the average. A valid PyQL can directly generate an executable SPARQL query. In detail, PyQL encapsulates various SPARQL syntax elements, such as Basic Graph Patterns, Assignments, Filters, Aggregations, and Subqueries. The detailed function list of PyQL can be found in Appendix H. The main features of PyQL can be summarized as follows:

- **User-friendly and conciseness**. PyQL offers an intuitive and concise approach for querying KB by shielding users from unreadable and lengthy database query language. It alleviates the burden of learning and writing SPARQL and effectively reduces the entry barrier for the community in utilizing SPARQL. It also makes sure that the generated SPARQL is grammatically correct and uniformly formatted. Specifically, in MarkQA, the average token length of PyQL is only 60.6% of SPARQL, and the grammar errors when using PyQL as output is half of those of SPARQL.

- **Step-by-Step reasoning path**. PyQL, in a symbolic manner, shows the transparent reasoning pathway of a question. Compared to SPARQL or S-expression, which is presented as a whole query and is difficult to parse or decompose, PyQL exhibits how to construct a query step by step. PyQL also serves as an efficient supervision signal, with our experiment showing up to a 19% performance improvement. With the prevalence of the Large Language Model (LLM) and Chain-of-Thought (CoT) (Wei et al., 2023), it is feasible to use

PyQL as a structural CoT. Besides, the code-style format also benefits LLMs in understanding due to their pre-training on code resources.

## 4 MarkQA

### 4.1 Dataset Construction

Our construction comprises 4 steps: Seeds collection, Paraphrasing, Generalization, and Composition, detailed in Figure 2. We extract this process into a general framework and name it **S**eeds-t**o**-**F**orest (SoF).

### 4.1.1 Seeds Collection

As mentioned in 2.1, we collect seed questions in an NLQ-to-LF manner. This allows for a greater diversity of questions, providing the possibility to reach questions with longer reasoning paths while ensuring the meaningfulness of each seed question. Besides, the question is more natural, as they are posed directly by humans, instead of transformed from randomly searched logic forms.

We invite 10 graduate students familiar with KBQA to annotate seed questions. We instruct them to focus on exploring patterns specific to certain relations, thus enhancing the variety and originality of the questions. Annotators are required to annotate at least three questions for each quantitative property and each question must involve at least one computational operation. Furthermore, for different questions related to the same relation, their computational structure must be different, which means that the differences between the two questions can not be merely the replacement of entities. In addition, all the entities and relations in the questions are recorded for logic form annotation purposes.

We then invite six graduate students familiar with QDMR, PyQL, and SPARQL to annotate the QDMR and PyQL for each seed question. We instruct annotators to first write down the QDMR (some sub-questions) of each question, then annotate the PyQL for each sub-question. The SPARQL of each question is automatically generated from PyQL and the annotators need to make sure that each SPARQL is executable with a unique answer. An annotated example and an auxiliary annotation system page can be found in Appendix D and E.

For each seed question, we ask another three annotators to check if it is meaningful and can be posed in the real world. We only keep the questions receiving over 2 approvals, resulting in 10.19%

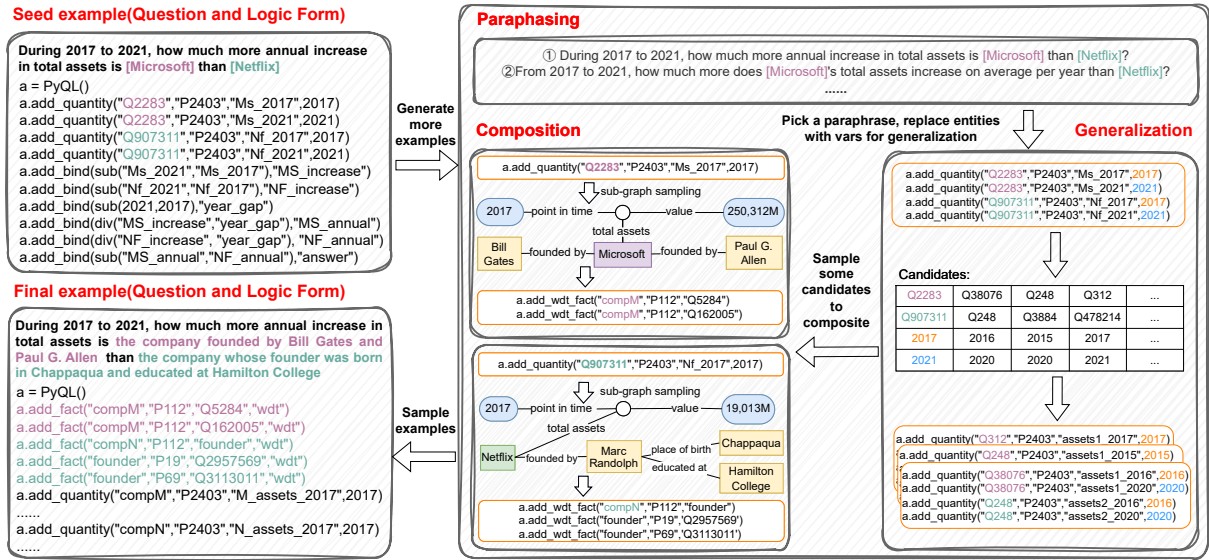

Figure 2: The Seeds-to-Forest (SoF) data construction framework. We first collect some seed questions and corresponding logic forms. And then generate more examples by paraphrasing, generalization, and composition.

dropped. For QDMR and PyQL, we ask another two annotators to check the correctness independently and correct all error cases. In the end, we collect a total of 950 seed questions for 318 quantitative properties.

### 4.1.2 Paraphrasing

We perform question paraphrasing through GPT-3.5-turbo to increase the surface-level diversity of the questions. The prompts and a paraphrased result can be found in Appendix F. Given a question, we enclose the mentioned entities in parentheses and anonymize them as A/B/C. Then, we ask the model to paraphrase the anonymized question, but when outputting, it should restore the specific entity labels based on the correspondence between A/B/C and the entity labels. In this way, we can obtain question paraphrases in a more controlled manner, and the model's output is directly usable. For each question, we instruct GPT-3.5-turbo to generate 10 paraphrases that preserve the original question's semantics.

We sample 1000 paraphrases of 100 questions and find only 4.3% with minor problems, so we consider the quality to be basically acceptable and obtain 9,366 paraphrases.

### 4.1.3 Generalization

In this step, we extract the logic form templates for each seed question by masking entities with variables. We execute the templates to acquire more entities that satisfy the SPARQL to generate more examples. In addition, for numerical literals in the

seed questions, we introduce some perturbations to avoid the model learning the bias from literals. We also collect the alias of the entity through the *skos:altLabel*. To ensure the quality and unambiguous nature of aliases, we have removed aliases that can be associated with multiple entities simultaneously, as well as aliases that are identical to the labels of other entities.

For each seed question, we sample up to 30 generalized questions. We employ the new entity's label or alias and its QID to replace the original in the question and logic form, respectively. As a consequence, we obtain about 20k examples.

### 4.1.4 Composition

So far, our emphasis has primarily been on exploring numerical reasoning. In real-world scenarios, numerical and multi-hop reasoning often arise together. In this step, we incorporate multi-hop reasoning into MarkQA by converting entities in the questions into descriptions. This is done through two steps: Sub-graph Sampling and Naturalization.

The aim of Sub-graph Sampling is to sample a sub-graph centering around the target entity for description. We consider structures within two hops, where each variable can be restricted by at most two triple patterns, resulting in six types of structures (detailed in Appendix J). These structures cover most structure patterns of previous complex KBQA datasets (e.g. LC-QuAD, CWQ, GrailQA). During sampling, we ensure that the sub-graph is sufficient (able to uniquely identify the entity) and non-

| Dataset | Size | NRQ | Canonical LF | Struct | SPT-OP | Avg NS |
|---|---|---|---|---|---|---|
| CWQ (Talmor and Berant, 2018) | 34,689 | 10.0% | 6236 | 56 | 7 | 0.30 |
| LC-QuAD 2.0 (Dubey et al., 2019) | 30,226 | 8.9% | 200 | 20 | 6 | 0.09 |
| GrailQA (Gu et al., 2021) | 51,100 | 16.2% | 4330 | 22 | 8 | 0.16 |
| KQA Pro (Cao et al., 2022) | 117,970 | 45.1% | 45563 | 418 | 6 | 0.57 |
| **MarkQA** | 31,902 | 100.0% | 12410 | 1054 | 15 | 2.51 |

Table 1: Detail statistics of MarkQA with other datasets. NRQ represents the percentage of questions that require numerical reasoning. Canonical LF means the number of distinct SPARQL templates, which is done by masking all entities, and types. Struct means the number of distinct skeleton structures of SPARQL, acquired by masking all entities, types, variables, and relations. We also took into consideration that different query graphs may be isomorphic heterogeneous graphs due to the different ordering of triples. SPT-OP refers to the number of numerical operations supported. Avg NS means the average number of computational operations.

redundant (without extra triple patterns). To guarantee the meaningfulness of the sub-graph, we collect high-quality relations from existing datasets, including SimpleQuestions, CWQ, and GrailQA. For Freebase relations, we transform them into wiki-dataIDs using Wikidata's official mapping page. We calculate their distribution as a base to select triple patterns and incorporate some smoothing techniques to balance the sampling probability and alleviate the long-tail phenomena.

For Naturalization, we prompt GPT-3.5-turbo to transform the sub-graphs into natural language descriptions. To prevent information leakage, we mask the target entity and the inner entity as a special token and instruct GPT-3.5-turbo which one to describe. In other words, it should not output the label of the target entity or the internal entity as a variable. Otherwise, it is not a real two-hop problem, since the internal variable is already leaked. The prompt we use can be found in Appendix G.

With our well-designed prompt, GPT-3.5-turbo excels at the task of converting sub-graphs into natural language descriptions. We sample 100 (sub-graph, description) pairs and find 93 acceptable. In this step, 50% of questions from 4.1.3 have one or two entities replaced with descriptions. Each question can generate at most 2 new questions through this step. Finally, we obtain 31,902 examples.

## 4.2 Dataset Analysis

### 4.2.1 Statistics

Our MarkQA consists of 31,902 examples. A detailed comparison with existing datasets is shown in Table 1. Compared with existing datasets, MarkQA has quite a lot more questions that require numerical reasoning (NRQ). In addition to being more numerous, MarkQA is superior in terms of

the difficulty of numerical reasoning (Avg NS) and supports more comprehensive operators (SPT-OP). Combined with multi-hop reasoning, the template of our query far exceeds others (Canonical LF). This results in more diverse questions that have not been included in previous datasets. Considering the query structure (Struc), MarkQA shows a great diversity compared to others. This is intuitive, as the combination of KB graph patterns and computational graphs produces a combinatorial explosion. We provide other detailed statistics in Appendix B.

### 4.2.2 Operators Analysis

The question of MarkQA considers 15 types of operators, including 5 arithmetic operations (addition, subtraction, multiply, divide, absolute value), 5 aggregation operations (count, argmin, argmax, average, summation), and 5 comparative operations($>$, $<$, $>=$, $<=$, $=$). With their combination, MarkQA supports most of the questions that require numerical reasoning. The percentage of questions that have at least one arithmetic, aggregation, or comparative operation are 88.1%, 24.8%, and 34.6%, respectively. There are 71.20% of questions with different operators and the average number of operators per question is 2.51.

### 4.2.3 Answer Analysis

The answers of MarkQA are all unique, which may manifest as a numerical value, an entity, or a boolean value. When presented with questions that produce collections of entities, we transform them into ones with unique answers (such as the size of the entity set or the maximum value of one of their attributes). The percentage of answers in the dataset that are a number, an entity, and a boolean value are 72.9%, 17.9%, and 9.1%, respectively.

| Methods | Output | Overall | I.I.D | Compositional | Zero-shot |
|---|---|---|---|---|---|
| **T5-base** | SPARQL | 34.24 | 70.05 | 53.71 | 6.32 |
| | PyQL | 40.70 | 78.32 | 63.10 | 10.39 |
| **GMT** | SPARQL | 38.68 | 78.32 | 63.58 | 6.07 |
| | PyQL | 43.63 | 82.10 | 68.33 | 11.71 |
| **QDTQA** | SPARQL | 37.19 | 76.82 | 57.37 | 7.01 |
| | PyQL | 42.57 | 84.59 | 70.89 | 7.01 |

Table 2: QA performance (%) on test set of MarkQA.

### 4.2.4 Quality

We invited three workers to evaluate the final dataset. Each example can be accepted if it receives over 2 approvals. In detail, we randomly sample 100 examples from the test set of MarkQA to examine their questions, QDMR, and PyQL. We find that all samples have fluent questions. 5 questions are ambiguous or not meaningful, which means they are not expected in real-world questions. The QDMR or PyQL of 8 questions is problematic. In total, 89 out of 100 examples are acceptable.

## 5 Experiment

### 5.1 Experimental Setup

Our training/validation/test sets contain about 70%/20%/10% of the data, corresponding to 22,352, 6,334 and 3,216 instances, respectively. Following (Gu et al., 2021), we evaluate the generalizability of MarkQA by three levels (i.e. I.I.D, Compositional, and Zero-shot). For the validation and test set, the proportion of the three levels of generalization remains the same as in the original paper (50% for Zero-shot, 25% for Compositional, 25% for I.I.D). Given that the answer of MarkQA are unique, we consider accuracy (ACC) as the evaluation metric.

### 5.2 Baselines

There are mainly two mainstream methods in KBQA, namely Information Retrieval (IR) methods and Semantic Parsing (SP) methods. IR methods retrieve a subgraph from KB and rank candidate entities in the subgraph to reach the answer. They can only handle questions whose answer is the entity in the sub-graph, unable to answer questions that require further reasoning (computation) of the nodes in multiple connected subgraphs. Therefore, we only consider SP methods, which parse natural language questions into executable

logic forms, as our baseline models. Concretely, we adapt T5-base and two representative SP methods to MarkQA: (1) **T5-base** (Raffel et al., 2020) is a language model, and we model the QA task as a sequence-to-sequence task. We concat question, entity linking, and relation linking results as input. (2) **GMT** (Hu et al., 2022) uses a multi-task learning framework to refine the retrieved linking results along with generating target logical forms. (3) **QDTQA** (Huang et al., 2023) improves KBQA performance by incorporating question decomposition information. For each model, we report the performance of two formats as output, i.e. SPARQL and PyQL.

### 5.3 Main Results

Table 2 summarizes the evaluation results on MarkQA. Compared with existing complex KBQA datasets, MarkQA is more challenging given that the performance of the state-of-the-art method drops dramatically. After involving PyQL, the overall performance improves significantly, where the relative increase ranges from 12.80% to 18.87%. It is easy to grasp since PyQL is a more concise representation and the average token length of PyQL is only 60.6% of SPARQL. Besides, the grammar of PyQL is simpler than SPARQL, resulting in fewer grammar errors. For 95% of the questions, the top 1 generated results are grammarly-valid when adopting PyQL as output, while this metric drops to 90% with SPARQL as the output, which indicates the superiority of PyQL.

The result also shows that Compositional and Zero-shot settings are challenging. We observe a maximum of 19.4% relative performance gap between the I.I.D and Compositional setting. Meanwhile, the Zero-shot setting is the most challenging among the three levels. It not only includes unseen schema items but also naturally contains unseen combinations of schema items which the Composi-

| Methods | Over. | I.I.D | Comp. | Zero. |
|---|---|---|---|---|
| **T5-base** | 40.7 | 78.3 | 63.1 | 10.4 |
| w/ gold E | 46.5 | 88.3 | 72.7 | 12.1 |
| w/ gold R | 479 | 79.2 | 65.5 | 23.1 |
| w/ gold ER | 57.6 | 89.8 | 76.1 | 31.9 |

Table 3: Detailed analysis of T5-base with PyQL as output. w/ gold E or R means we use golden entity or relation linking results. Over., Comp., and Zero. stands for Overall, Compositional, and Zero-shot, respectively.

| Type | Over. | I.I.D | Comp. | Zero. |
|---|---|---|---|---|
| All | 40.7 | 78.3 | 63.1 | 10.4 |
| NR | 42.0 | 85.4 | 64.3 | 12.9 |
| NR and MR | 38.5 | 65.5 | 61.8 | 5.1 |
| NR and MR(1) | 43.3 | 70.1 | 70.2 | 6.5 |
| NR and MR(2) | 28.7 | 55.6 | 44.8 | 2.3 |

Table 4: Performance of different types of questions on T5 (PyQL). NR and MR mean numerical reasoning and multi-hop reasoning, respectively. MR(1) and MR(2) mean one-hop and two-hop reasoning, respectively.

tional level focuses on. Moreover, some relations may have an entirely new query structure (masking all entities, types, variables, and relations) even after masking the relations. We also analyze the average PyQL length of the three settings and find no obvious differences among the three levels (from 7.4 to 7.8), indicating that the difficulty is not primarily brought about by the longer reasoning path.

### 5.4 Analysis

**Performance with golden linking** We perform an oracle experiment to find out why all models fail to achieve a satisfying performance and give an upper bound of these models on MarkQA. As shown in Table 3, given the golden linking results, the performance increases significantly on all settings, especially the Zero-shot setting with a 21-point increase. This meets the expectation as the Zero-shot setting additionally has unseen relations. However, even with golden linking results, the performance of all models is still poor. In specific, the performance of the Zero-shot setting is far behind the Compositional setting. It demonstrates that linking is indeed a challenge, but not the only challenge.

**Performance of different types of questions** Table 4 shows the performance of different types of questions. Row 1 represents the performance of all questions. Row 2 showcases the performance of a subset of questions that only requires numerical reasoning. By excluding newly generated questions in the Composition stage, this subset provides a measure of the inherent difficulty of numerical reasoning itself. The performance only sees a slight increase of 1.3%, indicating that numerical reasoning is indeed challenging and that the dataset's primary difficulty stems from this aspect. Row 3 displays the performance of questions that require both numerical and multi-hop reasoning. This subset of data is complementary to the data in Row 2. The performance drops by 2.2%, suggesting

that the inclusion of multi-hop reasoning further increases the question's difficulty. Rows 4/5 presents the performance of questions that require numerical reasoning and one-hop/two-hop reasoning, respectively. The questions with two-hop reasoning exhibit a significantly lower performance than the ones with one-hop reasoning.

## 6 Conclusion

In summary, our contributions are as follows:

1. We propose a new task NR-KBQA which necessitates the ability to perform both multi-hop reasoning and numerical reasoning.

2. We devise PyQL, a domain-specific language to portray the step-by-step reasoning process. Aside from providing convenience for annotation, it also has superiority when serving as a supervised signal for complex reasoning.

3. We construct a large-scale dataset, namely MarkQA, with 31,902 examples which is automatically scaled from 1K real questions. Each question is equipped with a QDMR, a PyQL, and a SPARQL.

4. We migrate two state-of-the-art baselines from Freebase to Wikidata. Experimental results show that NR-KBQA faces great challenges.

Overall, we for the first time explore and discuss numerical reasoning in KBQA from task formulation, reasoning paths representation, dataset construction, and baseline implementation. We believe this work should prompt the research in this area.

### Limitations

The major limitations of the work include: (a) The seed question is manually annotated. It would be more efficient to automatically collect

human-asked questions for constructing a large-scale dataset. (b) Our dataset construction process includes some synthetic steps, such as replacing the entity with a description, which may have resulted in some rigid expressions. (C) PyQL, for now, is SPARQL-oriented. It will be more universal after being generalized to other query languages, such as SQL, Lambda-DCS, and others.

## Acknowledgements

This work is supported by the National Natural Science Foundation of China (NSFC) under Grant No. 62072224. The authors would like to thank all the participants of this work and anonymous reviewers.

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

## A Numerical Properties Selection

We first select the Numerical Properties(num_p) with more than 100 statements or qualifiers recorded in Wikidata (of January 23, 2023). This leaves us with 377 num_p. This is to avoid selecting rare or long-tail numerical attributes since they are likely to be uncommon and difficult to comprehend. Additionally, if a numerical attribute has too few records, it becomes challenging to generate a sufficient number of questions during the generalization process, leading to data imbalance issues. Then, we manually remove some num_p requiring high-level domain knowledge (e.g. redshit, pKa, longitude of ascending node). Each removed num_p is checked by two annotators. If both annotators agree that it is hard to ask meaningful NRQ for this num_p, then this num_p is dropped. Finally, we retain 318 num_p.

## B Data Analysis

Here we provide more statistical details of MarkQA. Tabel 5 shows the number of distinct entities, properties, and answers covered by MarkQA, along with the average PyQL and question length. Table 6 presents the distribution of different numerical reasoning types among the three generalization levels. Tabel 8 shows the distribution of questions with different levels of the combination of numerical reasoning and multi-hop reasoning.

| Measurement | value |
|---|---|
| # distinct entity | 19,342 |
| # distinct property | 582 |
| # distinct quantitative property | 318 |
| # distinct answer | 9,677 |
| Avg PyQL len | 7.39 |
| Avg question length(tokens) | 25.6 |

Table 5: Key statistics for MarkQA

| Type | I.I.D | Comp. | Zero. | Over. |
|---|---|---|---|---|
| Arithmetic | 24.4 | 25.2 | 50.4 | 100.0 |
| Aggregation | 22.3 | 23.9 | 53.7 | 100.0 |
| Comparison | 27.8 | 33.9 | 38.3 | 100.0 |

Table 6: Distribution(%) of different levels with different numerical reasoning types. Note that there are overlaps among the three reasoning types (the sum of the percentages may not equal to 1).

| Reasoning type | Prop |
|---|---|
| Arithmetic | 88.1 |
| Aggregation | 24.8 |
| Comparison | 34.6 |
| Arithmetic and Aggregation | 14.3 |
| Arithmetic and Comparison | 28.6 |
| Aggregation and Comparison | 7.9 |
| Arithmetic and Aggregation and Comparison | 3.1 |

Table 7: Propertion(%) of questions with different numerical reasoning type and their combination.

| Type | Number of NRQ | Proportion |
|---|---|---|
| NR | 20,745 | 64.4% |
| NR and MR | 11,468 | 35.6% |
| NR and MR(1) | 7,606 | 23.6% |
| NR and MR(2) | 3,862 | 12.0% |

Table 8: Distribution of different types of question. NR and MR mean numerical reasoning and multi-hop reasoning, respectively. MR(1) and MR(2) mean one-hop and two-hop reasoning, respectively.

## C Experimental Details

### C.1 Knowledge Base

We utilize Wikidata-2023-01-23 dump as the knowledge base. This dump consists of 21 billion facts and takes 1.8TB uncompressed on disk, which is far larger than another popular used KB, i.e. Freebase with 8 billion facts. To lower the barrier of using Wikidata for KBQA, we elaborately tailor the Wikidata dump to get a subset that contains only QA-related facts. We remove the facts that include but are not limited to articles, references, multilingual, multimedia, tables, and others useless for QA. This leaves us with about 5 billion facts which is the same order of magnitude as Freebase.

### C.2 Implementation Details

For all baselines, we use the same linking results. For entity linking, we use open-source linking tools ELQ, to retrieve candidate entities, the corresponding scores, and the mentions. For relation linking, we use Falcon 2 (Sakor et al., 2020).

For GMT and QDTQA, we keep their original training settings. For T5-base, we adopt the implementation based on Hugging Face (Wolf et al., 2020), with Adafactor optimizer. The learning rates of all three models are set to 5e-5. All models are trained for 30 epochs. For evaluation, the beam size is 25. All models are trained and evaluated on

an Nvidia RTX3090 GPU.

### C.3 Error Analysis

We sample 100 error cases from GMT model in PyQL format and summarize the errors into the following categories: (1) Wrong reasoning path (28%): We manually analyze the percentage of questions that predicate a wrong reasoning path regardless of the linking result. (2) Entity linking error (42%) and Relation linking error (26%): We compare the KB item in the predicted PyQL with the golden ones and find that 68% of questions have at least one wrong item. (3) Grammar error (4%): This means the generated PyQL is grammatically incorrect and cannot be compiled to generate executable SPARQL.

# D Annotation Example

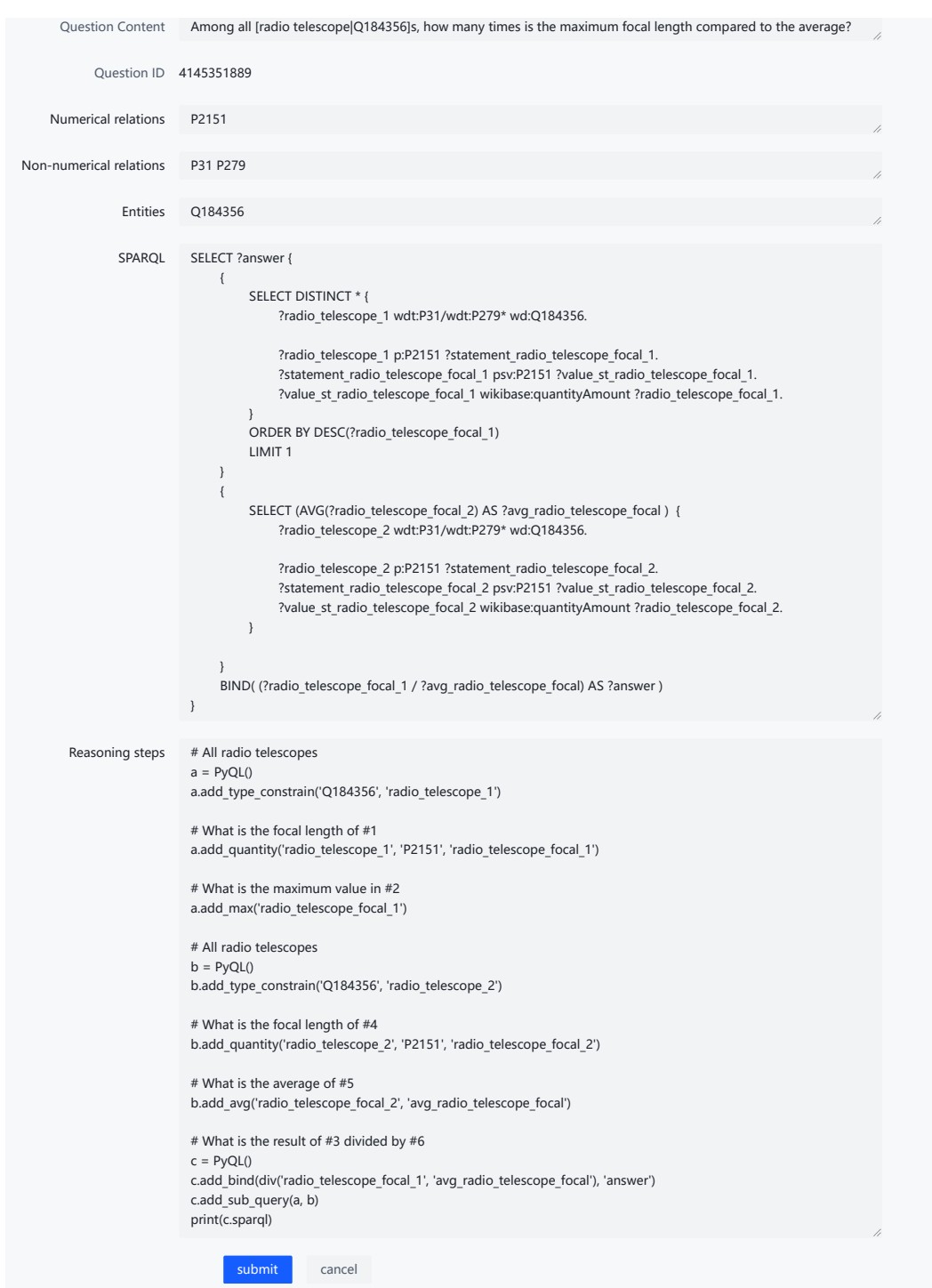

Figure 3: An annotation example of a seed question in our annotation system. Annotators are required to annotate the question, the ID of KB item (entities, numeral properties, non-numeral properties), SPARQL, and reasoning steps (QDMR and PyQL).

## E   Auxiliary Annotation System

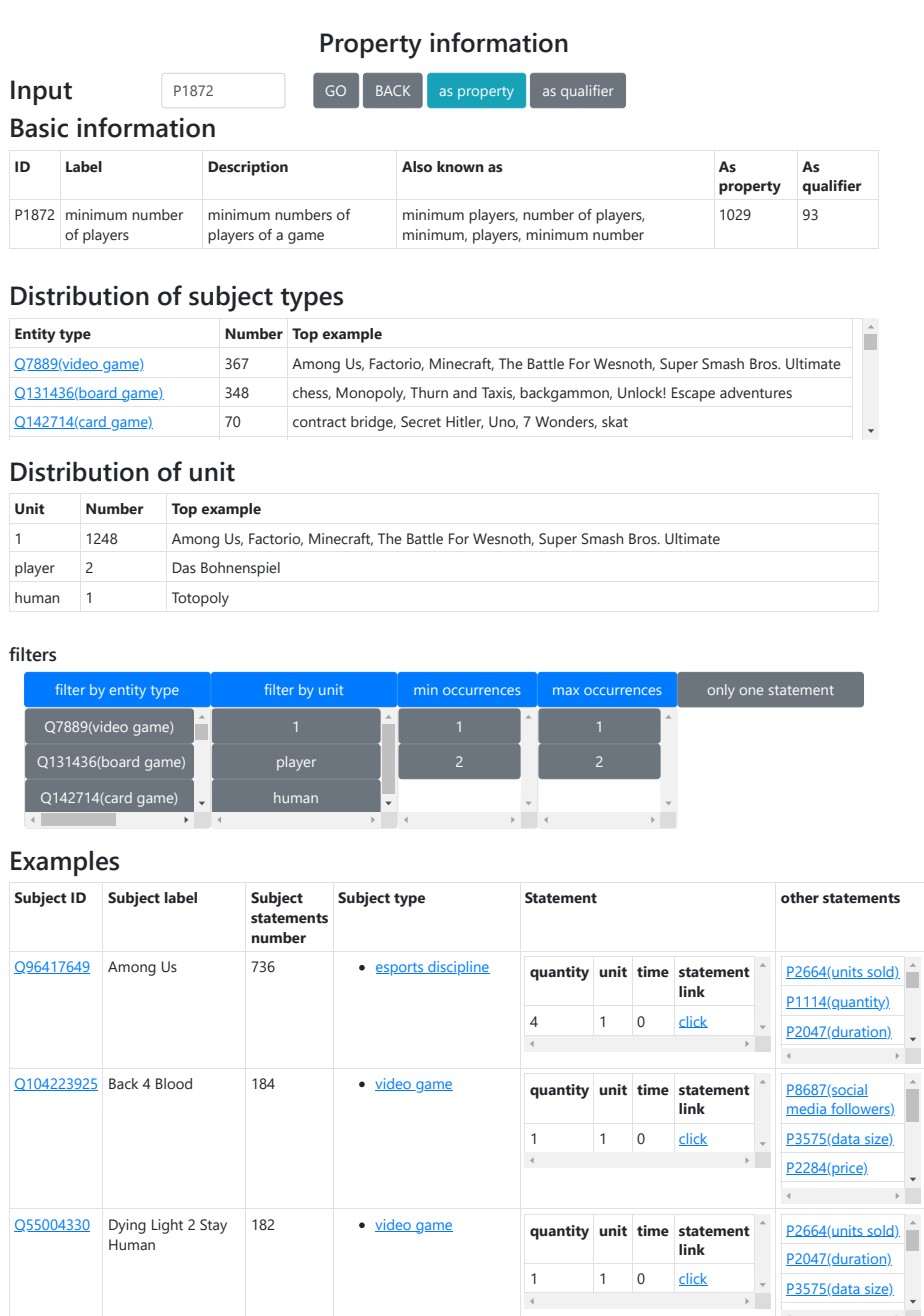

Figure 4: An example of our auxiliary annotation system. Given a quantitive property, the system provides comprehensive information including its label, aliases, description, subject type distributions, unit distributions of attribute values, and at most 2000 entities that have this property. Each entity is also provided with other quantitive properties the entity has to help the annotator ask questions with multiple quantity properties.

## F Prompt Used in paraphrasing

---

**Prompt used in paraphrasing**

Give 10 rewrites with the same meaning for each sentence following. Try to keep ALL the content within [] with the [] symbol in your output.

[]'s corresponding content is marked after the sentence. Use their meanings, but do not show them in rewrites.

For example: { [A]:[McDonald's] } . You know that [A] stands for McDonald's, but use [A] instead of McDonald's in rewrites.

So DO NOT contain []'s corresponding content in your output.

for example:

Among [A], [B] and [C], is the one with the largest load capacity also the most expensive to build? { [A]:MS Allure of the Seas [B]:Wonder of the Seas [C]:MS Oasis of the Seas }

a possible rewrite: Which of the three ships, [A], [B], or [C], has the highest cost of construction, and is also the one with the largest load capacity?

DO NOT contain []'s corresponding content in your rewrites!!

Output the output corresponding to each sentence in strict accordance with the following sample format:

sentence 1: ....

rewrite 1: ......

rewrite 2: ......

rewrite 3: ......

rewrite 4: ......

rewrite 5: ......

rewrite 6: ......

rewrite 7: ......

rewrite 8: ......

rewrite 9: ......

rewrite 10: ......

---

Table 9: Prompt used in paraphrasing. We request the model to perform a paraphrasing task and provide an example. Given a question, we enclose the mentioned entities in parentheses and anonymize them as A/B/C. This step is taken to prevent the model from omitting or tampering with the entity labels during the paraphrasing process. Then, we ask the model to paraphrase the anonymized question, but when outputting, it should restore the specific entity labels based on the correspondence between A/B/C and the entity labels. In this way, we can obtain question paraphrases in a more controlled manner, and the model's output is directly usable.

## G  Prompt Used in Composition

---

**Prompt used in composition**

Given some wikidata subgraphs centering on a center wikidata entity called [topic_entity], you need to get descriptions for each [topic_entity] base on their subgraph.

The input contains some subgraphs denoted by *subgraph_i*, with meanings of properties(*Property_Meanings*) in the end.
each *subgraph_i* contains:
[topic_entity]: the name of topic entity.
[type]: the most frequent wikidata type of topic entity.(some abstact entity may not have a type, then there is no [type] field.)
[triples]: some triples centered on [topic_entity]. A triples looks like <subject, property, object>.

The solution to this task is:
For each *subgraph_i*, use the semantics of its [triples] to get the description [topic_entity]. You need to include the meaning of all triples in the description.
A description is an appositive sentence of [topic_entity], but not contain the label of the topic entity.
To generate a description.............................
[ describing how to write a description]

ATTENTION:
1. PAY SPECIAL attention to directional properties ends with prepositions or has inverse property.......
2. The descriptions should be as short as possible(less than 6 tokens disregarding the entity label), but also need to be as diverse and fluid as possible. You may rewrite the meaning of properties.
..... [other attentions]

Examples of input and your output:
<input>
*subgraph_0*:{ [topic_entity]: Resurrection [type]:literary work [triples]:(<[topic_entity], writer, Leo Tolstoy>, <[topic_entity], genre, political fiction) }
*Property_Meanings*: (author:main creator(s) of a written work) (genre:creative work's genre or an artist's field of work (P101))
<endofinput>

[other 3 examples]
<output>
Solve1:
*subgraph_0* is a subgraph about Resurrection. Base on the meaning of triples,
Answer: |*subgraph_0*:[the @political fiction@ book written by @Leo Tolstoy@]|

[other 3 Solutions]
<endofoutput>
<endofprompt>

Now process following questions with subgraphs. Output strictly follows the example format.

---

Table 10: Prompt used in composition.

## H  Function List of PyQL

| PyQL function | Brief description |
|---|---|
| **Basic graph pattern** | |
| add_fact | Add a triple pattern. |
| add_quatity | Add a triple pattern with quantity relation. |
| add_quatity_with_qualifier | Add a triple pattern with quantity relation which is constrained by a qualifier. |
| add_quatity_by_qualifier | Add a qualifier with a quantity relation which is a constraint of a triple pattern. |
| add_type_constrain | Add a triple pattern with type constrain. |
| add_filter | Add a comparative constrain ($>$, $<$, $=$, etc). |
| add_bind | Bind the result of an expression to a variable |
| add_assignment | Bind some variables/entities to a variable |
| add_sub_query | Add a sub_query into current query |
| **Arithmetic** | |
| add | Addition of some numbers/variables. We also provide ceil and floor versions for add, sub, mul, and div. |
| sub | The difference between two numbers/variables. |
| mul | The product of two numbers/variables. |
| div | The division of two numbers/variables. |
| abs | The absolute value of a number/variable. |
| **Aggreggation** | |
| add_max | Calculate the maximum value of a given variable. Support to return the m-th latest to n-th largest item. |
| add_min | Calculate the minimum value of a given variable. |
| add_avg | Calculate the average value of a given variable. |
| add_sum | Calculate the summation value of a given variable. |
| add_count | Count the occurrences of matching triples or variables. |
| add_rank | Calculating the ranking of a variable in a list of variables. |
| **Boolean** | |
| add_compare | Whether two arguments meet a condition ($>$, $<$, $=$, etc). |
| **Other** | |
| add_sub_query | Add sub-queries to the outer query. |

Table 11: A brief introduction of PyQL's functions.

# I PyQL API implementation example

```python
class PyQL():
    def __init__(self):
        self.triple_pattern=[]
        self.aggregation=[]
        self.sub_query=[]
        self.head=''
        self.answer=''
        ...

    def add_type_constrain(self, type_id:str, new_var:str):
        self.add_triple_pattern(ent + " wdt:P31/wdt:P279* wd:" + type_id + ".")

    def add_count(self,count_obj:str,new_var:str, group_obj=None):
        if group_obj!=None:
            self.aggregation.append('GROUP BY '+group_obj)
            self.set_head('SELECT (COUNT(DISTINCT '+count_obj+') AS '+new_var+') '+' '+group_obj)
        else:
            self.set_head('SELECT (COUNT(DISTINCT '+count_obj+') AS '+new_var+') ')

    def add_max(self, max_obj:str, return_obj='*',offset=0,limit=1):
        self.aggregation.append("ORDER BY DESC(" + max_obj + ")")
        if limit!=None:
            self.aggregation.append("LIMIT "+str(limit))
        if offset!=0:
            self.aggregation.append("OFFSET "+str(offset))
        self.set_answer(return_obj)

    def add_bind(self, equation:str, var_name:str):
        self.add_triple_pattern('BIND( (' + equation + ') AS ' + var_name + ' )')
        self.set_answer(var_name)

    def add_filter(self, compare_obj1:Union[str,float], operator:str, compare_obj2:Union[str,float]):
        self.add_triple_pattern("FILTER(" + str(compare_obj1) +' '+ operator +' '+ str(compare_obj2)
            + ').')

    def add_assignment(self,var_list:list,new_var:str):
        self.add_triple_pattern('Values '+new_var+' {'+' '.join(var_list)+'}')

    def add_compare(self, obj1:Union[str,float], op:str, obj2:Union[str,float]):
        self.add_bind("IF(" + obj1 + " " + op + " " + obj2 + ', "TRUE", "FALSE")', "?answer")
        self.set_head("SELECT ?answer")

    def add_sub_query(self,*sub_query:PyQL):
        for sub_q in sub_query:
            sub_q = sub_q.sparql
            self.sub_query.append(sub_q)

    @property
    def sparql(self):
        self.triple_pattern_text = self.__construct_triple_pattern()
        self.sub_query_text = self.__construct_sub_query()
        self.aggregation_text = self.__construct_aggregation()

        if not self.head_already_set:
            self.head='SELECT DISTINCT '+self.answer

        sparql_temp = self.head + ' {\n' +
            self.sub_query_text + self.triple_pattern_text + '\n}\n'+ self.aggregation_text
        return sparql_temp

    ...
```

Figure 5: The implementation of some PyQL APIs. For the sake of brevity and better understanding, we have omitted some technical details.

# J Six Structure Considered in Composition

**One triple constrain target entity**
**Description**: the birthplace of Rudolf Riedlbauch
**Subgraph**: <Rudolf Riedlbauch, place of birth, [target_entity]>
([target_entity] is Dýšina)

**Two triple constrain target entity**
**Description**: the house located in Turtle Bay and designed by Perkins Eastman
**Subgraph**: <[target_entity], architect, Perkins Eastman>,
            <[target_entity], location, Turtle Bay>
([target_entity] is Turkish House)

**One triple constrain target entity, one triple constrain inner variable**
**Description**: the place where the child of Llewelyn Lloyd died
**Subgraph**: <[target_entity], child, [inner_variable]>,
            <[inner_variable], place of death, [target_entity]>
([target_entity] is Llewelyn Lloyd, [inner_variable] is Charles John Andersson)

**Two triple constrain target entity, one triple constrain inner variable**
**Description**:
the video game publisher whose founder was educated at Shanghai Jiao Tong University
**Subgraph**: <miHoYo, instance of, video game publisher>
            <miHoYo, founded by, [inner_variable]>
            <[inner variable], educated at ,Shanghai Jiao Tong University>
([target_entity] is miHoYo, [inner_variable] is Cai Haoyu)

**One triple constrain target entity, two triple constrain inner variable**
**Description**:
the director of the film received Academy Award for Best Picture and composed by Kris Bowers
**Subgraph**:<[inner_variable], director,[target_entity]>
            <[inner_variable], award received, film received Academy Award for Best Picture>
            <[inner_variable], composer, Kris Bowers>
([target_entity] is Peter Farrelly, [inner_variable] is Green Book)

**Two triple constrain target entity, two triple constrain inner variable**
**Description**: the World Cup Golden Ball winner who participated in the World Cup in Qatar
**Subgraph**: <[target_entity], award received, World Cup Golden Ball>
            < [target_entity], participant in, [inner_variable]>
            <[inner_variable], country, Qatar>
            <[inner_variable], sports season of league or competition, FIFA World Cup>
([target_entity] is Lionel Messi, [inner_variable] is 2022 FIFA World Cup)

Table 12: Six structures considered in composition.