# OpenReview forum: "MarkQA: A large scale KBQA dataset with numerical reasoning"
_EMNLP/2023/Conference — EMNLP 2023 Main_

### Official Review · Reviewer_Y7yn · 2023-08-04

**Soundness:** 4

**Excitement:**

3: Ambivalent: It has merits (e.g., it reports state-of-the-art results, the idea is nice), but there are key weaknesses (e.g., it describes incremental work), and it can significantly benefit from another round of revision. However, I won't object to accepting it if my co-reviewers champion it.

**Paper Topic And Main Contributions:**

In this paper, the authors propose a new challenging task, NR-KBQA, which focuses on numerical reasoning and its combination with multi-hop reasoning.

Specifically, they construct a large-scale dataset called MarkQA of 32K examples each with  QDMR  question decomposition and corresponding logic form in Python, i.e., PyQL.

PyQL provides a step-by-step reasoning process and can directly be compiled to executable SPARQL queries.

**Questions For The Authors:**

How do you obtain paraphrases in 4.1.2 Paraphrasing?

**Reasons To Accept:**

Numerical reasoning is a challenging and important problem in NLP. The authors propose a useful dataset that focuses on numerical reasoning, which can be a valuable testbed for future work.

**Reasons To Reject:**

1. Missing limitations.

2. A clear formulation of a sample in MarkQA should be described in a paragraph, referring to Figure 2 alone is not enough.

4. The experiments are not comprehensive. More baselines should be presented as a data resource paper. There are no ablation analyses for the numerical reasoning part.

**Reproducibility:**

4: Could mostly reproduce the results, but there may be some variation because of sample variance or minor variations in their interpretation of the protocol or method.

**Reviewer Confidence:**

4: Quite sure. I tried to check the important points carefully. It's unlikely, though conceivable, that I missed something that should affect my ratings.

---

> ### Author Rebuttal · Authors · 2023-08-29
>
> ### Q1: How do you obtain paraphrases in 4.1.2 Paraphrasing?
> Paraphrasing is obtained by gpt-3.5-turbo, we design a prompt to obtain paraphrase for each question. Details can be found in Appendix C.
>
> We request the model to perform a paraphrasing task and provide an example.
>
> Given a question, we enclose the mentioned entities in parentheses and anonymize them as A/B/C. This step is taken to prevent the model from omitting or tampering with the entity labels during the paraphrasing process.
>
> Then, we ask the model to paraphrase the anonymized question, but when outputting, it should restore the specific entity labels based on the correspondence between A/B/C and the entity labels.
>
> In this way, we can obtain question paraphrases in a more controlled manner, and the model's output is directly usable.
>
>
> ### Q2:  The experiments are not comprehensive and lack the ablation of numerical parts.
> Thanks for your suggestions, we promise to add the ablation study of the numerical part in the revised version.
> Here is a detailed analysis of numerical reasoning and multi-hop reasoning (The model is T5_PyQL).
>
> | Question type  |  Overall |  I.I.D |  Compositional  |  Zero-shot  |
> |---|---|---|---|---|
> |  All     |    43.7  |  84.1  |  67.9  |  10.9  |
> |  Only require numerical reasoning   | 45.4  |  92.4  |  69.6  |  13.3   |
> |  Both require numerical reasoning and multi-hop reasoning |  40.8  |  68.8  |  66.2  |  5.8  |
> |  Both require numerical reasoning and one-hop reasoning  |   45.7  |  74.3  |  75.2  |  6.5  |
> |  Both require numerical reasoning and two-hop reasoning  |  30.7  |  57.1  |  47.8  |  4.3 |
>
>
> Row 1 represents the overall performance of all questions.
>
> Row 2 showcases the performance of a subset of questions that **only** require numerical reasoning. By excluding newly generated questions in the composition stage, this subset provides a measure of the inherent difficulty of numerical reasoning itself. The performance only saw a slight increase of 1.7%, indicating that numerical reasoning is indeed challenging and that the dataset's primary difficulty stems from this aspect.
>
> Row 3 displays the performance of questions that require **both** numerical reasoning and multi-hop reasoning. This subset of data is complementary to the data in Row 2. The performance dropped by 2.9%, suggesting that the inclusion of multi-hop reasoning further increases the question's difficulty.
>
> Rows 4/5 present the performance of questions that require numerical reasoning and one-hop/two-hop reasoning, respectively.
>
> &nbsp;
>
> We also add another baseline (T5-large_PyQL). The corresponding accuracy is shown below.
>
> | Method  |  Overall |  I.I.D |  Compositional  |  Zero-shot  |
> |---|---|---|---|---|
> | T5-Large | 46.26 | 87.80 |  78.95 | 8.52 |

---

### Official Review · Reviewer_PuQT · 2023-08-04

**Soundness:** 3

**Excitement:**

3: Ambivalent: It has merits (e.g., it reports state-of-the-art results, the idea is nice), but there are key weaknesses (e.g., it describes incremental work), and it can significantly benefit from another round of revision. However, I won't object to accepting it if my co-reviewers champion it.

**Paper Topic And Main Contributions:**

The paper presents a KBQA dataset with the main focus of numerical reasoning (and also multi-hop). Authors argue that existing KBQA datasets do not contain sufficient numerical reasoning questions and hence there is a clear void in this space. Hence, they introduce MarkQA, a dataset that contains 100% numerical reasoning questions and answers.

Paper also presents PyQL, which is a logical form written in Python language. Authors claim that PyQL makes it easier for users to write compared to more formal and traditional SPARQL query language.  A PyQL can generate executable SPARQL queries and they have showed in evaluation, it is easier to generate through the learning algorithms.

**Questions For The Authors:**

Question A: Why don’t you consider this work as a resource paper? You mentioned in the introduction that no numerical reasoning questions exist for KBQA, but Table 1 shows other datasets that contain NRQ. Did you mean 100% only NRQ dataset? But then, can you frame your contribution as ‘you are introducing NRQ for KBQA’ as you mentioned in the introduction? This is contradictory and very confusing. Also, jeopardizing contribution claims.

**Reasons To Accept:**

•	Focuses on more challenging questions – NRQ in KBQA setting.

**Reasons To Reject:**

•	Main contribution claim of the paper is not clear. Please refer to questions section.

•	I wonder this paper should be in ‘resource’ track. Otherwise, only remaining contribution is PyQL format and its effect of NRQ, which is a very focused/narrow contribution.

**Reproducibility:**

3: Could reproduce the results with some difficulty. The settings of parameters are underspecified or subjectively determined; the training/evaluation data are not widely available.

**Reviewer Confidence:**

3: Pretty sure, but there's a chance I missed something. Although I have a good feel for this area in general, I did not carefully check the paper's details, e.g., the math, experimental design, or novelty.

---

> ### Author Rebuttal · Authors · 2023-08-29
>
> ### Q1: Why don’t you consider this work as a resource paper?
> We acknowledge that our main contribution is the MarkQA dataset, but we consider our work as a Question Answering track paper for 2 reasons:
>
> 1. In addition to the contribution of the dataset resource, this is the first work that thoroughly analyzes and discusses numerical reasoning questions over knowledge bases. To elaborate, we first formally model the task. To explore how to tackle this task, we discuss and provide reasoning paths as further resources. Also, we implement some baselines and showcase the usefulness of the reasoning paths.
>
> 2.  PyQL has shown its effectiveness for QA systems. This can also be seen as an improvement to the existing systems. What's more, in addition to NR-KBQA, we hope to showcase that representations like PyQL, designed through this scenario, can help advance the development of interpretable reasoning. More details can be referred to in Q3.
>
> ###　Q2: Can you frame your contribution as ‘you are introducing NRQ for KBQA’?
> Previous datasets, indeed, contain some questions with simple numerical reasoning.
>
> However, as we mentioned in the Introduction and Related Works, these datasets were not primarily focused on addressing numerical reasoning.
> The numerical questions in current datasets are far from being solved, both in terms of the number of numerical questions and the scenarios that their questions can support.
> None of them have provided a comprehensive resource for research from problem formulation, dataset resource, reasoning path representation, and baseline implementation.
>
> To promote the research in this area, we for the first time formulate this task and provide a high-quality, fully-annotated, and large-scale dataset.
> Based on your feedback, we will rephrase this statement to "We explore and discuss this issue for the first time." in the revised version to reduce controversy.
>
> ### Q3: PyQL is a very focused/narrow contribution.
>
> Regarding the significance of PyQL, its contributions to the community can be summarized as follows:
>
> 1. A transparent reasoning path. For a deep learning model, it is unknown how it obtains the output even though the answer is correct. If a reasoning path is offered, a model can directly learn how the answer is derived, improving the unexplainability.
>
> 2. Moreover, PyQL is a supervision signal, which is also a well-structured resource of reasoning, as the model can learn more efficiently with the symbolic form. For humans, an NRQ can be solved by a combination of retrieving and calculation steps. Given these steps, how to tackle a reasoning problem can be learned strategically. In addition, with the rise of CoT in LLMs, PyQL can be put into the few-shot prompts as a step-by-step reasoning demonstration. Also, the CoT itself is a supervision signal for the auto-regressive model and an interpretable path for human beings.
>
> 3. More readable, lower the threshold. Speaking of a SQL-like query, it is always time-consuming to write and revise when involving complex numerical operations like some aggregations. PyQL is more readable than SPARQL and can be converted to SPARQL non-destructively.  This provides a convenient resource for the KBQA community to write, check, and generate SPARQL. As mentioned in the Experiment, the average length of PyQL is only 57% of that of SPARQL.  After using PyQL, performance gains range from 33% to 69%.
>
> It is also worth noting that PyQL is not a specific design for our dataset. It supports most of the commonly used SPARQL grammar, including basic graph pattern, aggregation, sub-query, arithmetic, filter, and boolean. This is general enough to model most of the questions in previous datasets, not just ours. Besides, it is easy to extend the functionality through secondary development.
>
> Our aspiration of PyQL is to serve as a transparent reasoning pathway in the KBQA area, which can be used as a reasoning resource, not limited to a narrow usage for annotation or learning simplicity on our dataset.

---

### Official Review · Reviewer_6fJ7 · 2023-08-08

**Soundness:** 4

**Excitement:**

4: Strong: This paper deepens the understanding of some phenomenon or lowers the barriers to an existing research direction.

**Paper Topic And Main Contributions:**

This work proposes a new task NR-KBQA which focuses on complex numerical reasoning in the KBQA setting. To this end, they introduce a new dataset called MarkQA, composed of 1K 'seed' questions naturally created by human annotators, and scaled up to 32K examples using a mixture of automatic and verification processes. They also introduce an intermediate expression using PyQL, which aims to simplify the creation of a SPARQL query for those unfamiliar with it, and can be deterministically converted to such.

**Questions For The Authors:**

1. What portion of the dataset has entities replaced with 2-hop connections? If we discard these, how much of the dataset remains?
2. With gold Entity and Relation linking, what are the new numbers for any baseline you can compute on the portion of the dataset remaining? (Can you recompute table 4 last row with the reduced dataset?)

Based on these numbers, I'm willing to raise my score.

**Reasons To Accept:**

They offer a strong motivation in that other datasets or baselines offer only a limited evaluation of numerical questions in the KBQA setting, and offer a concrete resource that could be very useful for advancing the field in this direction. The paper is written well.

**Reasons To Reject:**

1. Key weakness, which is arguably subjective : I believe replacing any entity in the (already numerical question) with a 2-hop subgraph is adding complexity for complexity's sake, while simply being unrealistic.
Take the question in Fig 1 for example : "During 2017 to 2021, how much more annual increase in total assets is the company founded by Bill Gates and Paul G Allen than the company whose founder was born in Chappaqua and educated at Hamilton College"? This does not strike me as a question an actual human would ask.

Instead, you could simply ask the two sub-questions in any of the existing systems in literature, get Microsoft and Netflix respectively, then ask "During 2017 to 2021, how much more annual increase in total assets is Microsoft than Netflix?". I also think a 1-hop replacement directly in the question is reasonable. ("... total assets is company founded by Bill Gates than ...")

While I recognize this may seem arbitrary, there is a reason we don't see a lot of 4 or 5-hop questions in the general KBQA literature - it's unrealistic. To be clear, I think it's great that this work adds certain *numerical* complexities such as subtraction etc. But I'm not in favor of adding pointless complexity, and I'm current not sure how much of the poor performance of the baseline is being driven by such.

2. Lack of details about the numerical properties selected and more examples of numerical complexity not present in prior datasets would be useful.

**Reproducibility:**

3: Could reproduce the results with some difficulty. The settings of parameters are underspecified or subjectively determined; the training/evaluation data are not widely available.

**Reviewer Confidence:**

4: Quite sure. I tried to check the important points carefully. It's unlikely, though conceivable, that I missed something that should affect my ratings.

---

> ### Author Rebuttal · Authors · 2023-08-29
>
> Thank you for your valuable comments.
> ### Q1: What portion of the dataset has entities replaced with 2-hop connections? If we discard these, how much of the dataset remains?
> Here are some statistics of MarkQA.
>
> | Question Type | Number of Question |  Proportion |
> |---|---|---|
> | Questions only require numerical reasoning | 20745  | 64.4%  |
> | Questions with one-hop connections | 7606  |   23.6%  |
> | Questions with two-hop connections | 3862  | 12.0%  |
>
> Discarding the questions with two-hop connections, there remain 28351 questions accounting for 88.0%.
>
> ### Q2: With gold Entity and Relation linking, what are the new numbers for any baseline you can compute on the portion of the dataset remaining? (Can you recompute Table 4 last row with the reduced dataset?)
>
> Here lists the performance of the T5_PyQL after dropping the questions with two-hop connections:
>
> | Methods  |  Overall |  I.I.D |  Compositional  |  Zero-shot  |
> |---|---|---|---|---|
> | T5-base  |  45.5  |  87.5  |  71.8  |  11.7  |
> |  w/ gold E  |  49.6  |  94.8  |  77.0  |  13.7  |
> |  w/ gold R  |  56.0  |  89.2  |  75.3  |  30.0  |
> |  w/ gold ER  |  61.2  |  95.9  |  81.1  |  34.1  |
>
> Compared with the original, the performance of gold ER does not increase significantly (from 59.6 to 61.2) after discarding the two-hop questions. It indicates that the difficulty is primarily brought by numerical reasoning instead of multi-hop reasoning (To some extent, multi-hop reasoning has somehow been solved. The F1 score can achieve 94% on another multi-hop KBQA dataset, ComplexWebQuestion, by a T5 model given golden linking results (Hu et al., 2022).).
>
> > Hu, X., Wu, X., Shu, Y., & Qu, Y. (2022, October). Logical form generation via multi-task learning for complex question answering over knowledge bases. In Proceedings of the 29th International Conference on Computational Linguistics(pp. 1687-1696).
>
> &nbsp;
>
> **In addition to the questions you raised, please allow us to elaborate some necessary clarifications on other aspects.**
>
> &nbsp;
>
>
> ### 1. The overly complex question is unrealistic.
> It is worth noting that the difficulty of the example in Figure 1 stems from three factors:
>
> - Complex numerical reasoning,
>
> - Two-hop reasoning,
>
> - Two entities were replaced with different subgraphs.
>
> If a question does not meet all three conditions at the same time, it is likely to be considered acceptable.
>
> In our dataset, we have controlled the token length of the question to avoid the occurrence of overly complex and impractical questions.
> Therefore, a question that satisfies all three conditions is very rare (less than 1%) and this kind of question can be considered as representing the most complex structures in MarkQA.
> However, inevitably, some overly complex questions may still arise.
>
> The aim we posting this example is to better illustrate how numerical reasoning and multi-hop reasoning are intertwined, instead of encouraging extremely hard questions. We agree that it would be more readable to choose another more realistic question as an example.
>
> Another consideration is that, though difficult (and a little impractical), it is still a question that humans can solve.  Including a small subset of extremely hard questions enhances the robustness and compositional generalization capability of the model.
>
> ### 2. Lack of details about the numerical properties selected and more examples of numerical complexity not present in prior datasets.
>
> Thanks for your suggestions, we promise to add selection strategies of numerical properties in the appendix of the revised version and include more representative questions in the introduction.
>
> #### Details about the selection of numerical properties(num_p).
> We first select the num_p with more than 100 statements or qualifiers recorded in Wikidata (of January 13, 2023). This leaves us with 377 num_p.
> Then, we manually remove some num_p requiring high-level domain knowledge (e.g. redshit, pKa, longitude of ascending node). Each removed num_p  is checked by two annotators. If both annotators agree that it is hard to ask meaningful NRQ for this num_p, then this num_p is dropped. Finally, we retain 318 num_p.
>
> #### More examples of numerical complexity are not present in prior datasets.
> In terms of the number of operations, previous datasets involve numerical reasoning at most once (their #NRQ is nearly equal to #Avg NS in Table 1). And in terms of the kind of operations, they only consider the Superlative, COUNT, and Comparison. Their question is like:
>
> - Which lake has the largest catchment area? (GrailQA, requires 1 superlative operation)
> -  How many popular musics are influenced by the band which has ISNI 0000 0001 2369 4269? (KQA pro, requires 1 count operation)
> -  Which player started his career after 2004 and was the Cleveland Browns draft ? (CWQ, requires 1 comparison operation)
>
> However, many questions that are common in the real world are not supported by the previous dataset. Our dataset encompasses a broader range of numerical questions, such as:
> -  How many of Japan's largest sports stadiums could be filled with the number of new COVID-19 infections in Japan in 2023? (Require 3 operations: 1 summation, 1 superlative, and 1 division)
> -  How many times longer is the longest aircraft carrier than the shortest?  (Require 3 operations: 2 superlatives and 1 division)
> - How much more VAT do you have to pay to buy the most expensive iPhone 13 in Russia than in Japan? (Require in total of 4 calculations: 1 superlative, 2 multiplies, and 1 subtraction)

---

### Meta-Review · Area_Chair_KXtc · 2023-09-17

**Recommendation:** 4

**Metareview:**

The paper presents a KBQA dataset with the main focus of numerical reasoning (and also multi-hop). Authors showed limits of existing resources and propose a novel resource to address this issue.

---

### Decision · Program_Chairs · 2023-10-07

**Decision:**

Accept-Main

**Comment:**

The paper presents a KBQA dataset with the main focus of numerical reasoning (and also multi-hop). Authors showed limits of existing resources and propose a novel resource to address this issue.